# Genome-Wide Association Studies of Body Weight and Average Daily Gain in Chinese Dongliao Black Pigs

**DOI:** 10.3390/ijms26073453

**Published:** 2025-04-07

**Authors:** Min Huang, Wenyu Zhang, Jiangpeng Dong, Zhengyu Hu, Xuhui Tan, Hao Li, Kailing Sun, Ayong Zhao, Tao Huang

**Affiliations:** College of Animal Science and Technology & College of Veterinary Medicine, Zhejiang A&F University, Hangzhou 311300, China; minhuang0702@outlook.com (M.H.); wenyu1119@outlook.com (W.Z.); xuedongcan0529@163.com (J.D.); 17733388235@163.com (Z.H.); xuhui0122@outlook.com (X.T.); 15165646175@163.com (H.L.); sunkailing_kailyn@outlook.com (K.S.)

**Keywords:** Dongliao black pigs, genome-wide association studies, body weight, average daily gain, body mass index

## Abstract

In the domain of swine production, body weight (BW) and average daily gain (ADG) are recognized as the primary performance indicators. Nevertheless, the genetic architecture of ADG and BW in Dongliao black (DLB) pigs remains to be fully elucidated. In this study, we performed a genome-wide association analysis of BW, ADG, and body mass index (BMI) in 358 DLB pigs of different days of age. The genome-wide association study (GWAS) showed the following: (1) The most significant single nucleotide polymorphism (SNP) detected for BW was on *Sus scrofa* chromosome (SSC) 11:100,808 (*p*-value = 1.16 × 10^−6^) that was also the most significant SNP for ADG. (2) The most significant SNP associated with BMI was SSC17:51,463,521 (*p*-value = 5.16 × 10^−8^). (3) SNPs SSC10:6,523,844 and SSC17:23,852,682 were identified in both BW and ADG. A meta-analysis was conducted on BW at different days and demonstrated SSC5:39,028,335 (*p*-value = 8.37 × 10^−6^) which was not identified in the results of each single trait. The regions of two SNPs (SSC11:100,808, SSC4:10,703,277) exhibited considerable influence on both BW and ADG and the related regions were selected for linkage disequilibrium (LD) analyses that exhibited a notable linkage. In addition, several genes were identified that are associated with obesity and play roles in lipid metabolism, including *MACROD2*, *PHLPP2*, *CYP2E1*, and *STT3B*.

## 1. Introduction

Global pork consumption accounts for 31% of all meat in 2020 (https://www.fao.org/3/cb1993en/cb1993en_meat.pdf, accessed on 19 April 2024). In recent years, the demand for high-quality pork has been increasing [1]. The pig industry meets market demand with efficient meat productivity. Most of the economic traits of the pig have been significantly improved through continuous selection over time [2]. Body weight, a particularly important part of pig breeding, has received widespread attention [3]. Piglets with low birth weights are less able to suckle and do not have enough energy to withstand the cold, resulting in slow growth and high mortality [4]. The effect of average daily gain at different stages of pig productivity is an important indicator for assessing pig growth rate [5].

However, both body weight (BW) and average daily gain (ADG) are complex quantitative traits with moderately low heritability, regulated by multiple genes and influenced by genetic and environmental factors [6]. A study on the progeny of a cross between large white pig and European wild boar showed one quantitative trait locus (QTL) on *Sus scrofa* chromosome (SSC) 4 associated with growth traits [7]. The QTL associated with BW was found on multiple chromosomes in the progeny of a cross between Landrace and Korean native pigs [8]. Currently, abundant quantitative trait loci for BW and ADG have been identified in pigs. According to the animal QTL database (Release 53, https://www.animalgenome.org, accessed on 6 March 2025), among the 55,166 QTL from 819 publications found in pigs, a total of 637 and 926 QTL are associated with BW and ADG, respectively.

Genome-wide association study (GWAS) is a method for identifying variates that are associated with traits and exploring genetic ssvariation within the genome [9], which has been widely used in humans [10], plants [11], and animals [12]. The GWAS can be implemented with various statistical models, such as mixed linear models that incorporates one SNP at a time into the regression model to analyze the relationship between the marker and the target trait [13]. A number of studies determine genetic variation about BW and ADG in pigs through GWAS analysis. One study showed that seven SNPs that significantly associated with BW were identified on SSC 1, 3, 6, and 10 from Duroc × (Landrace × Yorkshire) hybrid pigs (DLY) [14]. The SNPs associated with ADG in Italian large white pigs were distributed on almost all chromosomes [5]. The GWAS result from progeny of Duroc × Erhualian crosses shown that *NDUFAF6*, *TNS1*, and *HMGA1* are candidate genes of BW and ADG [15]. Although lots of studies about pig growth traits have existed and this research provides assistance in elucidating the genetic basis of conformational traits in pigs, there is almost no GWAS study about BW and ADG in Dongliao black (DLB) pigs.

The DLB pig is one of famous indigenous breed originated from North China with a black coat color, compact and sturdy body, medium-sized head, slightly wide forehead, wide and straight back, stout hooves. The DLB pigs exhibit strong adaptability, being able to withstand cold weather and tolerate roughage. In this study, we performed a GWAS of BW and ADG traits in different ages in DLB based on the China Chip-1 porcine SNP50K Bead Chip. Our results identified genetic variants and candidate genes that were significantly associated with BW and ADG in DLB pigs.

## 2. Results

### 2.1. The Descriptive Statistics of the BW, ADG, and BMI Traits for DLB Pigs

In this study, 18 growth-related traits of DLB were recorded, including BW traits at 1, 26, 60, and 90 days of age, six ADG traits between different days of age, and eight body mass index (BMI) traits. The 1-day-old body weight (BW1) had the lowest coefficient of variation (CV) at 17.86% and 90-day-old body weight (BW90) had the highest at 34.21%, which increased with age. The ADG26–60 stage had an average rate of 114.09 g/d, while the ADG from 60 to 90 days of age (ADG60–90) had the fastest growth rate of 189.65 g/d. The CV of ADG gradually increased from 26.62% at ADG1–26 to 52.77% at ADG60–90. The mean values of BMI between different days of age increased gradually with age and the CV ranged from 12.52% to 22.37%. Heritability between different days of age ranged from 0.22 to 0.83, with the highest performance at 60 days of age. The ADG between different days of age showed high heritability, ranging from 0.72 to 0.97. The heritability of BMI was generally low, except for BMI1-BH (0.79) versus BMI90-BH (0.99) (Table 1).

### 2.2. Relationship Among the Traits of the BW, ADG, and BMI

The traits generally conformed to a normal distribution (Appendix A). We calculated correlations of phenotype and genetics between BW, ADG, and BMI traits, respectively (Figure 1). The results showed that both phenotypic and genetic correlation coefficients between BW1 and BW60 were less than 0.14. There was a genetically perfect positive correlation between BW26, 60, and 90 (Genetic correlation, *r_g_* = 1). Phenotypic and genetic correlations of ADG between different days of age were positively correlated, with correlation coefficients over 0.27. A genetically perfect positive correlation (*r_g_* = 1) was found between ADG from 1 to 90 days of age (ADG1–90) and ADG from 26 to 90 days of age (ADG26–90), as well as ADG1–90 and ADG60–90. Phenotypic correlations of BMI were lower between days of age than that of BMI calculated from body height at 60 (BMI60-BH) and 90 days of age (BMI90-BH) with highest correlations (Phenotypic correlations, *r_p_* = 0.59). Body mass index was calculated from body length at 1 day of age (BMI1-BL) and body mass index calculated from body length at 60 days of age (BMI60-BL); the body mass index calculated from body height at 1 day of age (BMI1-BH) was genetically perfectly negatively correlated (*r_g_* = −1), but perfectly positively correlated with all other BMIs (*r_g_* = 1). BMI was calculated from body height at 26 days of age (BMI26-BH) and BMI was calculated from body length at 90 days of age (BMI90-BL); BMI1-BH was genetically highly negatively correlated (*r_g_* < −0.9). The phenotypes of BW and ADG were mostly genetically negatively correlated, but not significantly. However, BW90 and ADG26–60 had the highest correlations, as well as BW90 and ADG60–90 (*r_g_* = −0.36). There was a complete positive genetic correlation between BMI1-BL, and BW1, BW60 (*r_g_* = 1), but there was a complete negative correlation with BW90 (*r_g_* = −1). Heritability was also higher between BMI26-BH and BW (*r_g_* > 0.58). For the phenotype correlation between ADG and BMI60-BH, BMI90-BH was high (*r_p_*, 0.46–0.84). Genetically, BMI1-BL was fully negatively correlated with ADG from 1 to 26 days of age (ADG1–26, *r_g_* = −1). BMI1-BL was fully positively correlated with ADGs at all days of age except for a non-significant genetic correlation with ADG60–90 (*r_g_* = 1). BMI26-BH was genetically fully negatively correlated with ADG60–90 (*r_g_* = −1) (Figure 1).

### 2.3. Genome-Wide Association Study of BW, ADG, and BMI in DLB Pigs

We performed GWAS analyses for these 18 traits. The results showed that Q-Q plots have no systematic inflation for all these traits (Appendix A). A total of 74 SNPs were detected in 18 traits in DLB at chromosome-wide significant levels (Figure 2). Among these SNPs, 6, 17, and 51 SNPs were significantly associated with BW, ADG, and BMI, of which, 30 annotated genes were obtained (Table 2). For the BW traits, six SNPs that were significantly associated with them were detected on SSC 3, 10, 11, and 17. The most significant SNP that located 13,517 bp downstream of ATP12A gene was on SSC11:100,808 with a *p*-value of 1.16 × 10^−6^ (Figure 2A). The analysis revealed a high level of significance for SNPs on SSC11 at the BW60, however, no significant SNPs identified for traits of BW1 and BW90. For the ADG, a total of 17 significant SNPs were detected on SSC7, 10, 11, and 17. The most significant SNP was SSC11:100,808 with a *p*-value of 2.10 × 10^−6^ (Figure 2B). This SNP was also detected in the BW60 trait and was the most significant SNP. The findings indicate that these SNPs exert a multitude of influences on disparate growth traits in DLB pigs. The GWAS results of ADG at different growth stages revealed that SNPs reaching statistical significance could be detected at the 60-day-old stage. However, no significant SNPs were detected at the 1-day-old and 90-day-old stages. For the BMI, 51 significant SNPs were detected on SSC 11. The most significant SNP was SSC17:51,463,521 with a *p*-value of 5.16 × 10^−8^. This SNP is the most significant SNP of all traits and is located on the annotated gene SLC9A8 (Figure 2C). For the different ages of BMI, it was observed that the significance SNP identified on SSC 4 was more favorable in BMI60-BH. However, a similar level of significance was not observed in other age groups.

### 2.4. Linkage Disequilibrium Block Analysis

To test the presence of interlocking among significant SNPs, we analyzed interlocking unbalanced blocks in the regions where the major SNPs were located. We analyzed linkage disequilibrium (LD) in the regions where the two major SNPs (SSC11:100,808, SSC4:10,703,277) are located. The SNPs of SSC11:82,720 and SSC11:100,808 are located in an LD block of length 18 kb (Figure 3A). The 16 kb Block1 contains SNPs of SSC4:10,703,277, SSC4:10,699,548, and SSC4:10,686,323 and has r^2^ = 1.0. The 2 kb of Block2 contains SNPs of SSC4:10,999,856 and SSC4:11,002,675 and has r^2^ = 1.0 (Figure 3B). Further, we analyzed the genotypic distribution of CNC10110003 (SSC11:100,808). The genotype GG individuals of DLB pigs exhibit significantly greater body weights than those with the AG or AA (Figure 3C,D). The individuals with genotype AA exhibited the lowest body weight and the slowest growth rate (Figure 3E,F). The DLB pigs with the GG genotype were observed to exhibit a higher BMI than pigs with the GA and AA genotypes (Appendix A).

## 3. Discussion

### 3.1. Trait Analysis of BW, ADG, and BMI of DLB Pigs

The CV of BW gradually increased with the age in pigs, indicating that individual differences gradually increased during growth [16]. The CV of BW for DLB pigs in our research was 25.39%, however, only 5.32% and 24.04% for Landrace and Yorkshire pigs [17], which may be caused by different breeds as well as rearing levels. A cross progeny from boars (Duroc × Saba) and sows (Yorkshire × [Landrace × Saba]) showed an ADG of 448 g/d at 30–60 days of age [18]. However, the ADG was 604.31 g/d in Duroc pigs [19]. In our study, the average daily weight gain of DLB pigs from 26–60 days of age was 114.09 g/d, and the fastest growing stage was from 60–90 days of age with an average ADG of 189.65 g/d. It is evident that DLB pigs grow more slowly in the early stages. The heritability of ADG in Duroc pigs was 0.26 [20], while it ranges from 0.72–0.97 in our results which may be caused by differences in breed, genotype, and feeding factors. In our study, the fastest growth rate was observed in the ADG60–90 stages, and the CV of ADG also increased gradually over time, indicating inter-individual growth differences increased and pigs grew faster in later stages. We found the low phenotypic and genetic CV between BW1 and BW60. In addition, the genetic correlations among BW26, 60, and 90 were complete positive. These results signify higher genetic correlation among BW at the older stage and evident genetic correlation of BW at the later stage of growth. Phenotypic and genetic correlations of ADG among different days of age were positive with high CV values. It is worth mentioning that our results are similar to others’ research [18], suggesting some consistency in ADG between different stages. In our study, the BMI showed low phenotypic correlations among different days of age, while high correlations between specific indicators, such as BMI60-BH and BMI90-BH, were found. Meanwhile, BW and ADG were mainly genetically negatively correlated, while BW90 had a high correlation with ADG26–60 and ADG60–90, which suggested a specific relationship between BW and ADG during the later stages. Our results also showed the complete positive genetic correlation between BMI1-BL with BW1 or BW60 and the complete negative correlation with BW90. In addition, the genetic correlations among BMI26-BH and BMI in different days of age were high. These results suggested a complex genetic relationship between BMI and BW. The high phenotypic correlation between ADG and BMI60-BH or BMI90-BH also suggests a strong phenotypic relationship between ADG and BMI.

### 3.2. Significant Loci and Quantitative Train Loci for BW, ADG, and BMI

We performed single-trait GWAS for BW, ADG, and BMI of DLB at 1, 26, 60, and 90 days of age. Seventy-four significant SNPs were identified on SSC 14. Six SNPs on SSC 3, 10, 11, and 17 were significantly associated with BW. One QTL with 40 kb in length significantly associated with BW on SSC11:33.02–33.06 Mb was identified in large white and Landrace pigs [17], while a region of SSC11:0.1–1.11 Mb was identified in this study. In fact, the QTLs related to BW were identified on several chromosomes, including 4, 6, 7, and 15 in White Duroc × Erhualian [21]. These SNPs on SSC7 significantly associated with BW are located at 27.28–35.07 Mb. Similarly, we identified one QTL on SSC7 (1.10–11.82 Mb) for ADG26–60 as well. Our results for BW significantly differed from previous studies, possibly due to differences in pig breeds.

For the ADG trait, a total of 17 significant SNPs were identified on SSC 7, 10, 11, and 17 in this study. Results of an ADG study on the progeny of the large white and crossbred sows of Landrace × large white showed that SNPs significantly associated with ADG were primarily located on SSC 2, 4, and 8 [22]. Interestingly, one SNP of SSC7:5,113,060 [5] that was near our results (SSC7:5,822,223) was found to be associated with ADG in Italian large white pigs. One QTL of SSC7:22.66–35.07 Mb significantly associated with ADG between days of age was identified in White Duroc × Erhualian F3 pigs [21]. Two SNPs significantly associated with ADG were identified (SSC7:113,231,192 and SSC10:448,069,639) in Duroc pigs [23]. Notably, SSC10:65,273,844 was detected simultaneously in BW26, ADG1–26. In addition, the same SNP of SSC17:23,852,682 was detected in traits of BW60, ADG from 1 to 60 days of age (ADG1–60), and ADG26–60. The results reflect the coherence of pig growth and the continuous role of genetic factors. This indicates that these loci have an important influence on pig growth from the early growth stage to the later stage. However, our results differed from previous studies which means that the QTLs affecting ADG in DLB pigs may be different from other pig breeds.

The BMI is defined as the ratio of weight to height [24]. The definition of height is different between humans and pigs. According to previous studies, the body length [25] and body height [26] of pigs can be used as “height” to calculate BMI. BMI is usually one of the body’s indicators for the degree of obesity [27]. Some studies have shown that Yorkshire pigs’ BMI-related loci are associated with obesity diseases [28]. Fifty-one SNPs significantly associated with BMI were identified on SSC 4, 6, 8, 9, 11, 13, 14, 15, 17, 18, and X in pigs. SNPs significantly associated with BMI were identified on SSC14:234,578 in crossbred pigs of Duroc × Landrace × Yorkshire [14], while 15 QTLs were found to be significantly associated with BMI on SSC 1, 2, 4, 7, 8, 9, 10, 17, and 18 in Yorkshire Pigs [29]. The QTLs associated with BMI are heavily distributed in different regions of multiple chromosomes, which means BMI is subject to micro-efficient polygenic control. The SNPs associated with BMI identified in this study differ from other studies, partly due to pig breed.

### 3.3. Candidate Genes Identified According to the GWAS Results

We identified 30 candidate genes associated with BW, ADG, and BMI. For BW, *MACROD2*, *ATP12A*, *FAM110C*, and *ASB13* were identified. The *MACROD2* gene has been reported to be associated with obesity in Koreans [30]. Carriers of the minor allele of one mutation of *MACROD2* reduced the risk of obesity and showed a trend toward a lower risk of BMI. Genes of *ASB13*, *ATP12A*, *MACROD2*, *SLC35B3*, *PDYN*, and *SIRPB2* were identified from ADG. Knockdown of *ASB13* promotes cell migration in breast cancer cells [31], and *ASB13* is associated with the development of acute myocardial infarction [32]. *ATP12A* is associated with the absorption and transport of water and electrolytes in the mouse gastrointestinal tract [33]. Also in the body, it is involved in the transportation of ions and water in the colon [34]. A total of 17 annotated genes including *PHLPP2*, *CYP2E1*, and *STT3B* were found by GWAS analysis of BMI. It has been shown that *PHLPP2* knockout mice exhibit enhanced systemic glucose tolerance and an increased rate of lipolysis [35]. The *CYP2E1* gene plays an up-regulatory role in lipid metabolism in the golden Jinhua pigs [36]. The *STT3B* gene is involved in fatty acid metabolism in Jining gray goats [37]. In addition, the *ATP12A* gene was identified from results of BW60 and ADG1–60 as well. The *MACROD2* gene was also identified from GWAS analysis for BW60, ADG1–60, and ADG26–60. Therefore, we consider *ATP12A* and *MACROD2* genes as strong candidate genes for BW and ADG in DLB pigs. Two genes identified in this study, *SLC9A8* and *KRR1*, are associated with BW traits in pigs. However, no other studies in the current study have reported that the *SLC9A8* and *KRR1* genes are associated with body weight traits in pigs. In summary, we identified several genes related to the growth traits of DLB pigs, some of which were found in other studies, and some were not reported. Our findings contribute new insights to the existing research on the genetic variation of porcine growth traits.

## 4. Materials and Methods

### 4.1. Animals and Phenotype

The experimental population consisted of 358 DLB pigs, including 294 females and 64 males. all raised under standardized conditions at the DLB pig farm (Shaoxing, Zhejiang, China). These pigs were derived from 118 distinct litters, ensuring genetic diversity and minimizing kinship bias. The farrowing dates were concentrated between March 2023 and July 2023. A total of four growth-related traits were obtained: body weights at 1-day-old (BW1), at the ages of 26-days-old (BW26), 60-days-old (BW60), and 90-days-old (BW90). The ADG was calculated between different days of age.BMI = weight/body length^2^

Body mass index (BMI) is the ratio of body weight to height squared, and body length and body height of DLB were used to calculate BMI. Weight is in kilograms and body length and height are in meters.

### 4.2. Descriptive Statistical Analysis

The traits generally conformed to a normal distribution. Phenotypic correlations (*r_p_*) between traits were estimated using the corrplot package in the R v4.3.2 [38]. Genetic correlation (*r_g_*) and heritability were estimated for each trait using GCTA v1.94 software [39]. The models and equations were as follows:

Bivariate REML model formula:$$\left[\begin{array}{c} y_{1}\\ y_{2} \end{array}\right]=\left[\begin{array}{cc} X_{1} & 0\\ 0 & X_{2} \end{array}\right]\left[\begin{array}{c} \beta_{1}\\ \beta_{2} \end{array}\right]+\left[\begin{array}{cc} Z_{1} & 0\\ 0 & Z_{2} \end{array}\right]\left[\begin{array}{c} u_{1}\\ u_{2} \end{array}\right]+\left[\begin{array}{c} \varepsilon_{1}\\ \varepsilon_{2} \end{array}\right]$$

Symbol definitions: $y_{1}$, $y_{2}$: Vectors of phenotypic observations for traits 1 and 2; $X_{1}$, $X_{2}$: fixed-effect design matrices; $\beta_{1}$, $\beta_{2}$: fixed-effects coefficient vectors; $Z_{1}$, $Z_{2}$: genetic random-effect design matrices; $u_{1}$, $u_{2}$: random genetic effect vectors $\left(u_{i}\sim N(0,G\sigma_{g_{i}}^{2})\right)$; $\varepsilon_{1}$, $\varepsilon_{2}$: residual error vectors $\left(\varepsilon_{i}\sim N(0,I\sigma_{e_{i}}^{2})\right)$.

Variance components:$$Var\left(\left[\begin{array}{c} u_{1}\\ u_{2} \end{array}\right]\right)=\left[\begin{array}{cc} \sigma_{g_{1}}^{2} & \sigma_{g_{12}}\\ \sigma_{g_{12}} & \sigma_{g_{2}}^{2} \end{array}\right]⊗G,Var\left(\left[\begin{array}{c} \varepsilon_{1}\\ \varepsilon_{2} \end{array}\right]\right)=\left[\begin{array}{cc} \sigma_{e_{1}}^{2} & 0\\ 0 & \sigma_{e_{2}}^{2} \end{array}\right]⊗I$$

Symbol definitions: $\sigma_{g_{1}}^{2}$, $\sigma_{g_{2}}^{2}$: additive genetic variance for traits 1 and 2; $\sigma_{e_{1}}^{2}$, $\sigma_{e_{2}}^{2}$: residual variance of traits 1 and 2; $\sigma_{g_{12}}$: genetic covariance between traits 1 and 2; ⊗: Kronecker product operator; G: genomic relationship matrix (quantifies genetic similarity between individuals); I: identity matrix.

Heritability estimation:$$h_{1}^{2}=\frac{\sigma_{g_{1}}^{2}}{\sigma_{g_{1}}^{2}+\sigma_{e_{1}}^{2}},h_{2}^{2}=\frac{\sigma_{g_{2}}^{2}}{\sigma_{g_{2}}^{2}+\sigma_{e_{2}}^{2}}$$

Symbol definitions: $h_{1}^{2}{,h}_{2}^{2}$: heritability of trait 1 and 2.

Genetic correlation:$$r_{g}=\frac{\sigma_{g_{12}}}{\sqrt{\sigma_{g_{1}}^{2}\cdot \sigma_{g_{2}}^{2}}}$$

Symbol definitions: $r_{g}$: genetic correlation between traits.

### 4.3. Genotyping and Quality Control

Total DNA was extracted from these 358 pigs and genotyped using the China Chip-1 porcine SNP50K Bead Chip (http://www.kangpusen.com/breeding/8.html, accessed on 6 March 2025), and 57,466 SNP markers across the genome were obtained. Quality control was performed using PLINK v1.90 [40]. The filtering criteria were (1) individual call rates > 0.9, (2) SNP call rates > 0.9, (3) minor allele frequencies > 0.01, (4) *p* > 10^−6^ for Hardy–Weinberg equilibrium test, (5) SNPs with no position information were removed. Finally, a total of 358 pigs with 50,739 SNPs were retained for subsequent analysis.

### 4.4. Genome-Wide Association Studies

The GWAS was performed using GEMMA V.0.98 through a linear mixed model (LMM) [41]. The model is specified as follows:*y = Wα + xβ + µ + ε*
where *y* denotes a vector of phenotypic values; *W* refers to the incidence matrices of covariates (fixed effects), including sex; *α* is the vector of corresponding coefficients with the intercept; *X* represents the vector of SNP genotypes; *β* stands for the corresponding effect size of the SNP; *u* is the vector of random effects with *u*~*MVNn* (*0*, *λ τ*^−1^*K*); ε corresponds to the vector of random residuals with *ε*~*MVNn* (*0*, *τ*^−1^*In*); *λ* specifies the ratio between two variance components; *τ*^−1^ signifies the variance of the residual errors; *K* represents a genomic relatedness matrix between individuals was estimated via GEMMA; *I* refers to an *n × n* identity matrix, and *n* is the number of individuals; *MVNn* is the n-dimensional multivariate normal distribution. The genome-wide significant (0.05/N) and suggestive (1/N) thresholds by Bonferroni correction, in which N is the number of SNPs, were used in the analysis.

### 4.5. Linkage Disequilibrium Analysis

The linkage disequilibrium (LD) blocks were identified in the chromosomal regions containing the identified significantly associated SNPs using the software Haploview v4.2 [42]. The LD blocks were defined using Haploview based on default parameters according to the criteria.

### 4.6. Candidate Genes Identification

The positional information of the significant SNP was compared with Sus_scrofa.Sscrofa11.1.111.gtf. The gene with the closest distance to the SNP was used as a candidate gene. The position of the SNP with the candidate gene was confirmed by Ensembl database [43].

## 5. Conclusions

In conclusion, correlations between traits can serve as a guide for breeding efforts in DLB pigs. These correlations indicate which traits should be prioritized in the breeding process. In the case of DLB pigs, highly correlated traits can be strengthened concurrently in the breeding process to maximize the efficiency of genetic progress. By leveraging these correlations, the most advantageous and complementary cross combinations for DLB pigs can be identified. The GWAS results for the three traits BW, ADG, and BMI showed a range of genetic variants and candidate genes. For example, *MACROD2*, *PHLPP2*, *CYP2E1*, and *STT3B* are associated with obesity in the body, play a role in fat metabolism, and there is a close link between them and growth traits. The results of this study reveal the complexity of the genetic structure of growth traits in pigs.

## Figures and Tables

**Figure 1 ijms-26-03453-f001:**
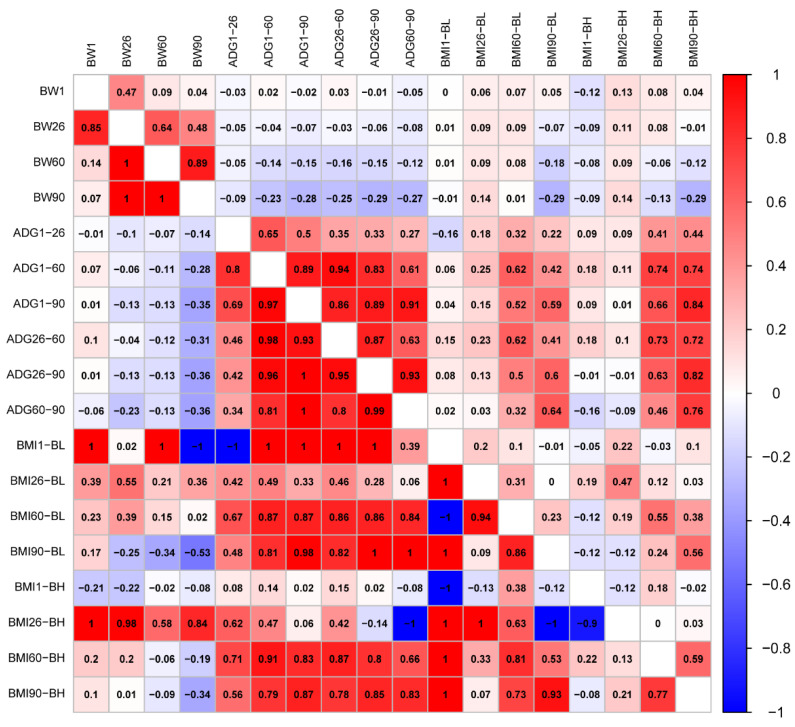
Phenotypic correlations are shown above the diagonal. Below the diagonal is the genetic correlation.

**Figure 2 ijms-26-03453-f002:**
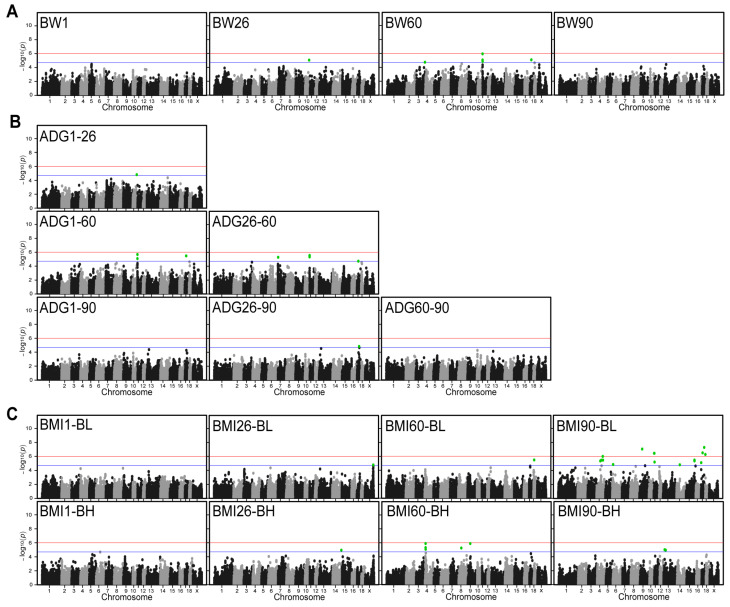
Manhattan plots for genome-wide association analysis of the BW, ADG, and BMI. (**A**) Manhattan plots of weight traits at different days of age. (**B**) Manhattan plots of average daily body weight gain between different days of age. (**C**) Manhattan plots of BMI at different days of age. *x*-axis indicates chromosomes and *y*-axis indicates −log10 (*p*–value). The blue and red horizontal lines indicate the thresholds for suggestively (*p*–value = 1.97 × 10^−5^) and genome-wide (*p*–value = 9.85 × 10^−7^) significant SNPs, respectively. Green dots represent SNPs that reached the significance threshold.

**Figure 3 ijms-26-03453-f003:**
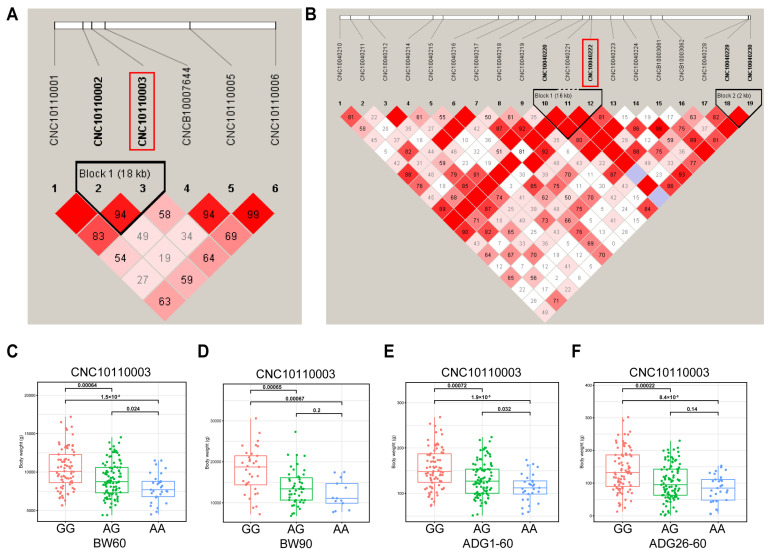
The LD block region for the identified SNP. The LD blocks are colored in accordance with the standard Haploview color scheme: LOD > 2 and D’ = 1, red; LOD < 2 and D’ = 1, blue; LOD < 2 and D’ < 1, white (LOD is the log of the likelihood odds ratio, a measure of confidence in the value of D’). (**A**) The LD block analysis of the 18-kb region that surrounds the significant SNP of SSC11:100,808 for BW60. (**B**) The LD block analysis for the region identified by SNP of SSC4:10,703,277 for BMI60-BH. (**C**–**F**) The association between CNC10010003 (SSC11:100,808) and phenotypes of BW60, BW90, ADG1–60, and ADG26–90.

**Table 1 ijms-26-03453-t001:** Variation of body weight conformation traits in DLB.

Trait	N	Mean	SD	Min	Max	CV (%)	h^2^ ± SE
BW1 (g)	358	1279.66	228.01	730	1860	17.86	0.37 ± 0.26
BW26 (g)	358	5486.26	1193.97	2190	8635	21.76	0.22 ± 0.21
BW60 (g)	208	9461.76	2529.21	4400	17,200	26.73	0.83 ± 0.19
BW90 (g)	92	15,046.12	5146.90	6830	30,615	34.21	0.68 ± 0.22
ADG1–26 (g)	358	168.26	44.80	55.00	276.40	26.62	0.72 ± 0.22
ADG1–60 (g)	208	138.31	42.41	51.53	268.31	30.66	0.96 ± 0.17
ADG1–90 (g)	92	154.89	57.78	66.01	325.45	37.30	0.97 ± 0.17
ADG26–60 (g)	208	114.09	60.85	1.18	302.06	53.34	0.83 ± 0.19
ADG26–90 (g)	92	146.20	73.69	15.25	354.14	50.41	0.94 ± 0.18
ADG60–90 (g)	92	189.65	100.08	8.33	503.83	52.77	0.81 ± 0.22
BMI1-BL (kg/m^2^)	358	25.74	3.22	18.51	40.59	12.52	0.01 ± 0.18
BMI26-BL (kg/m^2^)	358	35.24	5.38	20.23	55.85	15.28	0.24 ± 0.23
BMI60-BL (kg/m^2^)	208	36.42	5.34	21.53	52.48	14.66	0.19 ± 0.21
BMI90-BL (kg/m^2^)	92	39.81	8.07	27.32	93.97	20.27	0.49 ± 0.21
BMI1-BH (kg/m^2^)	358	44.10	8.84	24.03	79.56	20.03	0.79 ± 0.20
BMI26-BH (kg/m^2^)	358	79.30	14.35	48.21	122.31	18.10	0.03 ± 0.18
BMI60-BH (kg/m^2^)	208	95.51	15.46	38.25	138.67	16.19	0.62 ± 0.21
BMI90-BH (kg/m^2^)	92	105.56	23.61	55.75	154.79	22.37	0.99 ± 0.16

SD: standard deviation. CV: coefficient of variation. h^2^ ± SE: Heritability ± standard error. BW1: 1-day-old body weight; BW26: 26-day-old body weight; BW60: 60-day-old body weight; BW90: 90-day-old body weight. ADG1–26: average daily gain from 1 to 26 days of age; ADG1–60: average daily gain from 1 to 60 days of age; ADG1–90: average daily gain from 1 to 90 days of age; ADG26–60: average daily gain from 26 to 60 days of age; ADG26–90: average daily gain from 26 to 90 days of age; ADG60–90: average daily gain from 60 to 90 days of age. BMI1-BL: body mass index calculated from body weight and body length at 1 day of age; BMI26-BL: body mass index calculated from weight and body length at 26 days of age; BMI60-BL: body mass index calculated from weight and body length at 60 days of age; BMI90-BL: body mass index calculated from weight and body length at 90 days of age; BMI1-BH: body mass index calculated from weight and body height at 1 days of age; BMI26-BH: body mass index calculated from weight and body height at 26 days of age; BMI60-BH: body mass index calculated from weight and body height at 60 days of age; BMI90-BH: body mass index calculated from weight and body height at 90 days of age.

**Table 2 ijms-26-03453-t002:** Significant loci for BW, ADG, and BMI traits identified by China Chip-1 porcine SNP50K Bead Chip based genome-wide association studies.

Trait	Chr	N_snpsuggest_	Pos (bp)	*p*-Value	CI (Mb)	Alleles	Effect	Gene	Distance (bp)
BW26	10	1	65,273,844	9.05 × 10^−6^	59.27–65.27	A/G	−946.24	*ASB13*	−1688
BW60	3	1	137,139,318	1.85 × 10^−5^	131.14–138.55	A/G	1220.47	*FAM110C*	4,322,417
BW60	11	3	100,808	1.16 × 10^−6^	0.10–1.11	A/G	−1254.58	*ATP12A*	−13,517
BW60	17	1	23,852,682	8.35 × 10^−6^	23.85–29.85	C/T	1972.73	*MACROD2*	0
ADG1–26	10	1	65,273,844	1.54 × 10^−5^	59.27–65.27	A/G	−34.45	*ASB13*	−1688
ADG1–60	11	2	100,808	2.10 × 10^−6^	0.10–1.11	A/G	−20.67	*ATP12A*	−13,517
ADG1–60	17	1	23,852,682	3.33 × 10^−6^	23.85–23.85	C/T	34.54	*MACROD2*	0
ADG26–60	7	1	5,822,223	5.14 × 10^−6^	1.10–11.82	C/T	48.78	*SLC35B3*	215,692
ADG26–60	11	2	129,686	2.80 × 10^−6^	0.10–6.13	A/G	30.14	*ATP12A*	0
ADG26–60	17	1	23,852,682	1.88 × 10^−5^	23.85–29.85	C/T	45.72	*MACROD2*	0
ADG26–90	17	3	33,471,951	1.44 × 10^−5^	27.47–39.47	A/G	46.25	*PDYN*	−33,128
ADG26–90	17	3	33,497,319	1.44 × 10^−5^	27.50–39.50	C/T	46.25	*PDYN*	−7760
ADG26–90	17	3	33,737,738	1.44 × 10^−5^	27.74–39.74	A/G	46.25	*SIRPB2*	−120,582
BMI26-BL	X	2	136,625,261	1.65 × 10^−5^	130.63–137.33	A/G	1.78	*-*	-
BMI26-BL	X	2	136,628,078	1.65 × 10^−5^	130.63–137.33	G/T	1.78	*-*	-
BMI60-BL	18	1	12,824,362	3.27 × 10^−6^	6.82–12.82	C/T	−5.30	*CHRM2*	224,176
BMI90-BL	4	5	101,009,461	9.82 × 10^−7^	95.01–104.03	A/G	19.47	*NOTCH2*	0
BMI90-BL	6	1	14,648,172	1.43 × 10^−5^	8.65–20.65	A/G	9.74	*PHLPP2*	0
BMI90-BL	9	1	45,103,299	8.85 × 10^−8^	45.10–45.10	G/T	28.91	*DSCAML1*	0
BMI90-BL	11	7	7,065,106	3.57 × 10^−7^	7.07–13.07	C/T	26.76	*KATNAL1*	667
BMI90-BL	11	7	8,057,101	3.57 × 10^−7^	7.07–14.06	G/T	26.76	*B3GLCT*	150,052
BMI90-BL	11	7	8,577,181	3.57 × 10^−7^	7.07–14.58	C/T	26.76	*ZAR1L*	−218,306
BMI90-BL	14	2	53,641,877	1.56 × 10^−5^	53.64–53.67	A/T	19.74	*RYR2*	−10,608
BMI90-BL	14	2	53,669,348	1.56 × 10^−5^	53.64–53.67	G/T	19.74	*RYR2*	0
BMI90-BL	15	2	137,743,389	3.28 × 10^−6^	137.04–143.74	G/T	15.68	*ILKAP*	0
BMI90-BL	17	3	51,463,521	5.16 × 10^−8^	45.46–51.46	A/G	13.56	*SLC9A8*	0
BMI90-BL	18	1	1,401,569	5.28 × 10^−7^	1.40–1.40	C/T	19.58	*PTPRN2*	11,962
BMI26-BH	14	1	142,385,942	1.13 × 10^−5^	136.39–146.61	A/G	6.82	*CYP2E1*	649,125
BMI60-BH	4	3	10,703,277	1.26 × 10^−6^	6.15–13.89	C/T	9.41	*ASAP1*	251,348
BMI60-BH	8	1	42,799,214	5.62 × 10^−6^	42.80–48.80	G/T	10.41	*TLL1*	−57,255
BMI60-BH	9	1	39,683,032	1.29 × 10^−6^	39.68–43.94	A/G	−7.56	*C11orf52*	27,809
BMI90-BH	13	2	17,568,893	3.27 × 10^−6^	1.40–1.40	A/G	17.07	*STT3B*	0

Chr: The chromosomal location of the most significant SNPs. N_snpsuggest_: the number of SNPs that reached the genome significant threshold. Effect: the effect size of the marker in the formula. Nearest gene: The nearest annotated genes from the most significant SNPs. Distance (bp): The distance from the most significant SNPs to the nearest genes. Minus indicates SNP locates on the upstream of target gene, positive indicates SNP locates on the downstream of target gene, zero indicates SNP locates in the gene.

## Data Availability

The data used in this study are available at a publicly available repository https://www.jianguoyun.com/p/DSwT31UQ_YGCDRjCpOgFIAA (accessed on 6 March 2025).

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
