# Peer review of "Genome-Wide Association Studies of Body Weight and Average Daily Gain in Chinese Dongliao Black Pigs"

_ijms, 2025, doi:10.3390/ijms26073453_

Round 1

Reviewer 1 Report

Comments and Suggestions for Authors

In this paper, the author used the GWAS to identify some SNPs related to BW, ADG, and BMI in a pig population. This is an interesting work. Nevertheless, some issues still need to be further addressed.

1.Line 44: the author should clearly specify the version of the database

2.Line 71-77: The author's description of the studied population is overly simplistic. What is the composition of the studied population? This is unclear to the reader. How many families or lineages do these populations come from, or are they random populations? The author should explain this clearly. What is the level of feeding and management for this studied population, this is also not explained by the author. Is this studied population comprised entirely of sows or boars? 

3.Line 75: What is the unit of body length? The author should specify.

4. Line81: please fixed the software version

5.Line87:Please explain the reason for selecting a minor allele frequency of 0.01? Why not 0.05? Given this, I am concerned about the reliability of the GWAS analysis results.

6.Line87: please fixed the error for 10-6

7.Line92: hybrid model or mixed model? please explain it.

8.Line 110: Please describe in detail the process of candidate gene mining. The current description does not provide me with useful information.

9. In Table 1, for the indicators of average daily weight gain, the data shows that they all have sufficient coefficients of variation? How did the author handle this, and an explanation should be provided. I am very skeptical about the reliability of the heritability results.

10.Line103: “chromosome-wide significance level is 1/SNPs”, please explain it. Is there any literature to support your point of view?

Reviewer 2 Report

Comments and Suggestions for Authors

Dear authors, here are sveral remarks for improving the paper:

-line 121: in the results chapter, you mentioned heritability. I recommend that you give an information how did you estimate it in MM. The same is for coreelations, line 142. 

Conclusion should be expanded with practical meaning of this results, how they could be used in the developement of the breed. 
